# Human Mutant Dynactin Subunit 1 Causes Profound Motor Neuron Disease Consistent with Possible Mechanisms Involving Axonopathy, Mitochondriopathy, Protein Nitration, and T-Cell-Mediated Cytolysis

**DOI:** 10.3390/biom15121637

**Published:** 2025-11-21

**Authors:** Victor Xie, Maria Clara Franco, Lee J. Martin

**Affiliations:** 1Department of Pathology, Neuropathology Division, Johns Hopkins University School of Medicine, Baltimore, MD 21205, USA; vxie1@jhu.edu; 2Center for Translation Science, Herbert Wertheim College of Medicine, Florida International University, Port St. Lucie, FL 33199, USA; marfranc@fiu.edu; 3Department of Cellular & Molecular Medicine, Herbert Wertheim College of Medicine, Florida International University, Port St. Lucie, FL 33199, USA; 4Department of Neuroscience, Johns Hopkins University School of Medicine, Baltimore, MD 21205, USA; 5Department of Anesthesiology & Critical Care Medicine, Johns Hopkins University School of Medicine, Baltimore, MD 21205, USA; 6Pathobiology Graduate Training Program, Johns Hopkins University School of Medicine, Baltimore, MD 21205, USA

**Keywords:** ALS, cell death, CD8, heat-shock protein-90, Fas, mitochondria, motor neuron, T cell

## Abstract

Mutations in the gene encoding the p150 subunit of the dynactin complex (*DCTN1*) are linked to amyotrophic lateral sclerosis, spinal and bulbar muscular atrophy, and Perry syndrome. These neurodegenerative diseases can cause muscle weakness and atrophy, parkinsonian-like symptoms, and paralysis. To examine the evolution of neuropathology caused by a mutation in *DCTN1* and cellular mechanisms of disease for therapeutic discovery, we characterized mice expressing either human wildtype or mutant (G59S) DCTN1. Neuron-specific expression of mutant, but not wildtype, DCTN1 caused fatal age-related paralytic disease and motor neuron (MN) degeneration in the spinal cord with axonopathy and chromatolysis without apoptotic morphology. MNs became positive for cleaved caspase-3, cleaved caspase-8, and nitrated Hsp90. Mitochondria accumulated and appeared fragmented and dysmorphic and then were lost. This pathology was accompanied by invasion of CD95- and CD8-positive mononuclear T cells into the ventral horn and accumulation of TNFα and IL9. Administration of the mitochondrial division inhibitor-1 (Mdivi-1) protected MNs and extended the lifespan of G59S-*DCTN1* mice. A mitochondrial permeability transition pore inhibitor also extended lifespan. Thus, mutant *DCTN1* causes degeneration of MNs associated with axonopathy, mitochondriopathy, nitrative stress, and caspase activation. It appears as retrograde neurodegeneration and inflammatory T-cell-like cytolysis. Mitochondria are possible therapeutic targets in *DCTN1*-linked neurodegenerative disorders.

## 1. Introduction

Missense mutations in the dynactin gene (*DCTN1*) encoding dynactin subunit-1 are linked to several neurodegenerative diseases. *DCTN1* mutations can cause familial and sporadic amyotrophic lateral sclerosis (ALS) [1,2,3] and other neurodegenerative diseases manifesting as Perry syndrome [4]. The DCTN1 protein is the p150 component of a multi-protein complex that drives fast retrograde transport of vesicles, organelles such as mitochondria, RNAs, and proteins along microtubules [5,6,7]. DCTN1 co-purifies with dynein and is suggested to enhance the processivity and efficiency of the dynein motor [8,9]. In vitro studies demonstrate that dynein–dynactin complexes exhibit bidirectional processive motility [10]. The disruption of the dynactin complex by overexpression of dynamitin and by chemically induced mutations in dynein causes motor neuron diseases in mice [9]. However, the exact pathogenic mechanisms driving this neurodegeneration, as well as the diverse DCTN1-related neuropathologies associated with Perry syndrome, are unknown. Dynactin-microtubule abnormalities might involve cellular autophagy [11], nuclear motility [12], retrograde transport [13], protein nitration disruption [14], and protein aggregation [15]. Indeed, neurons could be exquisitely vulnerable to perturbations in retrograde transport of cargo, including mitochondria, from the distal axon [16,17]. Neuronal retrograde axonal transport is necessary to suppress cell death signals and neurodegeneration [18]. Thus, mutant DCTN1 could activate a form of retrograde neurodegeneration akin to that seen with axotomy [19,20,21].

Animal models could be useful for determining the mechanisms whereby human mutant DCTN1 causes clinical disease and degeneration of neurons. We used transgenic (tg) mice expressing either human wildtype DCTN1 or mutant G59S-DCTN1 driven by the excellent neuron-specific Thy1.2 promoter [22,23,24]. To isolate the potential cell autonomous effects of mutant DCTN1 involving axonopathy and retrograde neurodegeneration, neuron-restricted expression of human p150 was used rather than a more cell-generic promoter. These mice develop robust clinical and neuropathological phenotypes that are consistent with ALS, including significant loss of motor neurons in the spinal cord [23]. Here, we show that these mice develop severe axonopathy, T cell infiltrations, tumor necrosis factor-α (TNFα) and interleukin-9 (IL9) upregulation, and motor neuron degeneration. The process involves mitochondrial abnormalities, heat-shock protein-90 (Hsp90) nitration, prominent caspase activation, and DNA fragmentation. Although motor neurons acquired DNA double-strand breaks and there was the presence of caspase-3 and caspase-8 activation, the degeneration did not achieve the morphological finality consistent with apoptosis. Rather, it appeared as cell lysis attenuated by the mitochondrial division inhibitor Mdivi-1. Both Mdivi-1 and the mitochondrial permeability transition pore inhibitor GNX-4728 extended the lifespan of DCTN1-G59S tg mice. Thus, mitochondria are a possible therapeutic target in DCTN1-linked human neurodegenerative diseases.

## 2. Material and Methods

### 2.1. Ethics Statement

All animal experimentation was conducted in accordance with the ARRIVE guidelines and the guidelines of the Helsinki Declaration and was approved by the Institutional Review Board of Johns Hopkins University (protocol number MO22M353, approval 10/24/22).

### 2.2. Tg Mice and Treatments

Tg mice expressing either wildtype or mutant DCTN1 driven by the neuron-specific Thy1.2 promoter were generated originally as described [23]. Encoded *DCTN1* gene products in humans and mice have similar mobilities on SDS-PAGE, and no species-specific antibodies are available; therefore, to distinguish human DCTN1 from endogenous mouse DCTN1, full-length human DCTN1 cDNAs encoding wildtype or mutant variants were epitope-tagged with FLAG at the carboxyl termini. This was performed as described using a PCR/oligonucleotide primer-directed mutagenesis strategy [23]. Previous work has shown that the Thy1.2 cassette works well for generation of mice with neuron-specific expression of transgenes [22,24,25]; thus, expression constructs encoding human wildtype DCTN1 (Thy1.2p150WT) or mutant G59S-DCTN1 (Thy1.2p150G59S) were subcloned into the Xho I site of the mouse Thy1.2 expression cassette. The purified inserts from these vectors were submitted to the Transgenic Core Facility at Johns Hopkins University School of Medicine for injection into C57BL/6; SJL hybrid mouse embryos. Founder mouse lines expressing human wildtype and mutant DCTN1 were identified by Southern blotting [23]. Subsequent generations and additional mice (strain # 017590) obtained from the Jackson Laboratory (The Jackson Laboratory, Bar Harbor, ME, USA) were also identified by Southern blot analysis and using PCR analysis of the tail snips. The primer sets were CAA ATG GGT AGG CGT GAT TCT and GTC CCC GCA GTT TGC TAG TC to identify the human *DCTN1* and ATG GGT GGG CGT GAT TCT G and ACT GGC GTA CAA AGA TGC CG to identify the endogenous mouse *DCTN1*.

To clearly visualize the presence of degenerating axons, wildtype DCTN1 and mutant G59S-DCTN1 tg mice were crossed to tg mice expressing yellow fluorescent protein (YFP) under promoter control of regulatory elements derived from the mouse Thy1 gene for selective expression of YFP only in neurons and their axons [22,24,25]. Breeder mice (B6.Cg-Tg;Thy1-YFP-16Jrs/J, strain # 003709) were purchased from the Jackson Laboratory.

Additional tg mice were used to examine mechanisms of disease related to mitochondria. Wildtype and mutant DCTN1 mice (*n* = 6/genotype) were treated at 4 months of age with bromodeoxyuridine (BrdU) to study mitochondrial DNA (mtDNA) replication as a surrogate for mitochondrial division because it is an DNA S-phase marker [26]. BrdU (Sigma, St. Louis, MO, USA) was injected (50 mg/kg, intraperitoneally in normal saline) once, and the mice were killed 12 h later [27]. Spinal cord sections were processed as described [27]. Other groups of mutant DCTN1 (*n* = 12/group) mice received craniotomies and were surgically implanted at 4 months of age with single intracerebroventricular metal cannulas (secured by dental acrylic) into the right lateral ventricle (bregma −0.22, 1 lateral, 1.3 ventral) [28] and were infused with Mdivi-1 (MedChemExpress, Monmouth Junction, NJ, USA; 2 µL of 25 mg/mL) or vehicle (saline/β-cyclodextrin/DMSO) every other day for 4 weeks. Injection volume was delivered by slow infusion over 30 min. Half of the vehicle- and Mdivi-1-treated mice were killed at 8 months of age, and the other mice survived to endstage disease. In a pilot drug-placebo therapeutic study, other mutant DCTN1 mice (*n* = 8/group) were systemically treated (300 µg in 100 µL, intraperitoneally) with the mitochondrial permeability pore inhibitor GNX-4728 [29] or its vehicle (saline/β-cyclodextrin/DMSO) every other day for 3 months starting at 4 months of age. Mice survived endstage disease.

### 2.3. Characterization of Clinical Disease

Mice were regularly monitored to assess the onset and progression of symptoms associated with transgene expression. Observations were made regarding the presence of tremors, abnormality of gait or balance (ataxia), dystonia, spasticity, and ability to groom as well as weight. Overall motor activity was quantified by nocturnal running wheel assessments [30,31]. Mice were considered at endstage of disease when they were unable to right themselves within 10 s when placed on their back.

### 2.4. Histology and Neuropathology

Mice were killed by carbon dioxide inhalation and were rapidly perfused through the heart with ice-cold phosphate-buffered saline (PBS, pH 7.4) and then 4% paraformaldehyde in PBS (pH 7.4). The spinal cord and right sciatic nerve were removed and postfixed in the same fixative and then cryoprotected in 20% glycerol–PBS. Serial transverse sections of the L4-L5 lumbar spinal cord were cut on a sliding microtome (40 μm) and stained with cresyl violet. Other sections were stored in sodium acetate antifreeze buffer (−20 °C) until they were used. L3 lumbar spinal cord segments and transverse segments of the sciatic nerve were postfixed in 2% glutaraldehyde and processed for Epon plastic embedding as described [32]. Semithin 1 µm thick sections were cut on glass knives, attached to glass microscope slides by heating, and stained with toluidine blue for degenerating axon counting.

Cresyl-violet-stained spinal cord sections were analyzed systematically by light microscopy at 200× magnification to estimate segmental total motor neuron number and motor neuron degeneration. Motor neurons were identified by their large soma size and prominent nucleus. Neurons were considered intact when they had a well-defined cell body and a preserved Nissl substance and showed no evidence of cytoplasmic vacuolization or cellular fragmentation or dissolution. They were classified as damaged when exhibiting morphological irregularities such as cytoplasmic vacuolization, chromatolysis, karyolysis, or nucleus eccentricity. Cells with pale cytoplasm and indistinct nuclear boundaries were also classified as damaged.

Cell death in the ventral horn of the spinal cord was analyzed using terminal deoxynucleotidyl transferase dUTP-nick end labeling (TUNEL) to detect DNA double-strand breaks [21] as described [33] but using a free-floating section protocol [34]. Diaminobenzidine (DAB) was used as the chromogen. Positive cells with DNA fragmentation had brown nuclear labeling.

For degenerating axon counting in plastic sections, only myelinated axon cross-sectional profiles, mostly Aα and Aβ fibers, were counted at 1000× magnification. Small-diameter nonmyelinated or lightly myelinated Aδ fibers were not counted. Normal axons had a darkly stained myelinated ring around a microtubule- and mitochondrial-containing axoplasm. Degenerating axons had abnormal myelin whorls, dysmorphic shapes, and a darkly stained core axoplasm.

### 2.5. Immunohistochemistry

Spinal cord sections were removed from antifreeze storage buffer and rinsed thoroughly in tris-buffered saline (TBS) or PBS and immunohistochemically stained as free-floating sections for protein localization using the peroxidase anti-peroxidase (PAP) method [35,36] with DAB as the chromogen. Spinal cord sections from WT-DCTN1 and G59S-DCTN1 mice were always stained concurrently so that the sections were always exposed to identical solutions for the same time. Human dynactin presence was confirmed by staining for FLAG (M2 mouse monoclonal, Sigma) to distinguish human p150 from mouse p150. To immunophenotype the neurodegeneration, adjacent spinal cord sections were stained for cleaved caspase-8 (Cell Signaling Technology, Danvers, MA, USA), cleaved caspase-3 (Cell Signaling Technology), SOD2 (StressMarq Biosciences, Victoria, BC, Canada), Parkin-E3 ubiquitin-protein ligase (Proteintech, Rosemount, IL, USA), and nitrated Hsp90 [37]. To identify small round mononuclear non-neuronal cells that appeared as inflammatory cells that marginated degenerating motor neurons, spinal cord sections were immunostained for Fas (CD-95) and the cytotoxic T cell marker CD8. Inflammatory cytokines were visualized with antibodies to tumor necrosis factor-α (TNF-α) and interleukin-9 (IL9). BrdU incorporation into mtDNA was detected with monoclonal antibody [27]. The sections were blocked and permeabilized (1 h) in 10% normal goat serum (NGS) with 0.4% Triton-x 100 and then incubated (4 °C) in primary IgG antibodies overnight: rabbit monoclonal anti-cleaved caspase-8 (1:500, D5B2), rabbit monoclonal anti-cleaved caspase-3 (1:4000, D3E9), rabbit polyclonal anti-SOD2 (1:2000), rabbit polyclonal anti-Parkin (1:500), rabbit polyclonal anti-mouse TNF-α (Chemicon, St. Louis, MO, USA, 1:500), rabbit monoclonal anti-IL9 (Abcam, Cambridge, MA, USA, 1:500), monoclonal anti-nitrated Hsp90 (1:1000), a mouse monoclonal anti-Fas (BD Transduction Laboratories, Franklin Lakes, NJ, USA, 1:500), and a rabbit polyclonal antibody to CD8 (Proteintech, 1:500). Negative control sections were incubated with an equivalent concentration of purified non-immune mouse or rabbit IgG. Sections were then rinsed in TBS and then incubated (45 min) in secondary antibody (1:200, AffiniPure Goat Anti-Rabbit IgG or Goat Anti-Mouse IgG, Jackson ImmunoResearch Laboratories, West Grove, PA, USA). Sections were then rinsed in TBS and incubated (45 min) in peroxidase-conjugated tertiary antibody (1:400 rabbit or mouse PAP, Jackson ImmunoResearch Laboratories). Afterwards, the sections were rinsed again in TBS and reacted with hydrogen peroxide and DAB to visualize immunoreactivity.

The immunohistochemical staining of spinal cord sections was quantified by profile counting of individual cells and mitochondria within cells. Cleaved caspase-3 (CC3), cleaved caspase-8 (CC8), and nitrated Hsp90 immunopositive large cells were counted in the ventral horns at 400× magnification. Cells were classified as positive if they exhibited DAB staining above the faint amorphous diffuse background/negative control staining and had the appearance of a biological cell or organelle clearly divisible from artifact or debris. Tissue section storage solution contained sodium azide to prevent growth of debris. All solutions used were sterile-filtered to maintain a high level of immunohistochemistry quality control. Motor neurons lacking detectable DAB staining were classified as negative. Fas-positive cells in the ventral horn were counted at 400× magnification. These were small (<8 µm in diameter), round-nucleated cells generally with strong immunoreactivity. The SOD2 staining of mitochondria was quantified in individual motor neurons at 1000× magnification as described using filar micrometry [34]. This method entailed careful incremental focusing in the *z*-axis. SOD2-profiles that were <0.5 µm were ranked small, and those >0.5 µm were ranked large.

### 2.6. SDS-PAGE and Immunoblotting

Protein extracts from mouse unfixed frozen CNS tissue were homogenized in 1:10 PBS containing 1% SDS and protein inhibitor cocktail (Roche, Burlington, MA, USA). Samples were sonicated prior to protein assay and protein blot analysis. A total of 20 µg of total protein extracts was loaded onto 4–12% NuPAGE precast gels (Invitrogen, Waltham, MA, USA) and transferred to nitrocellulose membranes (Invitrogen). Immunoblotting was carried out using monoclonal mouse primary antibodies to actin (Millipore, St. Louis, MO, USA) and FLAG (M2, Sigma). FLAG was used because there are no specific antibodies to distinguish human dynactin p150 from mouse dynactin p150. Immunoreactivity was visualized using enhanced chemiluminescence.

### 2.7. Data Presentation and Statistical Analyses

The data were analyzed using GraphPad Prism 9.5.1 or XLSTAT 2023.1.5 software. Data normality assessments were performed using the Shapiro–Wilk test. There was no exclusion of data points for any data set. Motor activity, cell counts, and mitochondrial quantifications were analyzed by one-way ANOVA and a Holm–Sidak post hoc test. Statistical outcomes with this approach were compared to those determined by Student’s *t* test. Results are presented as box and whisker plots with mean and interquartile range (IQR). Mouse group sizes for experimental designs and data presentation were *n* = 6 unless otherwise indicated. The level of significance was set at *p* < 0.05.

## 3. Results

### 3.1. Verification of Tg Mice Expressing Human DCTN1

The expression of human wildtype and mutant transgenes in mouse CNS was confirmed by PCR and assessments of FLAG-tagged DCTN1 in spinal cords of tg and non-tg mice (Appendix A). It was favorable to use FLAG-tagged DCTN1 because no human-specific DCTN1 antibodies are available. Using a highly specific epitope-tag monoclonal antibody to FLAG, Western blotting identified a FLAG-tagged protein with a molecular weight consistent with the mobility of DCTN1 [38] in tg mouse CNS but not in non-tg mouse CNS (Appendix A). Immunohistochemistry using FLAG antibody revealed directly that the human wildtype and mutant DCTN1 were present in spinal cord motor neurons of these mice (Appendix A). FLAG immunoreactivity was not detected in non-tg mouse spinal cord sections (Appendix A).

### 3.2. G59S-DCTN1 Tg Mice Develop a Tremorous, Disequilibrium Clumsy and Spastic Phenotype That Evolves into Fatal Paralysis

G59S-DCTN1 and WT-DCTN1 mice were born with normal motor behavior and coordination. They showed initial clinical signs of neurological disease with a low-to-the-ground walking posture, shortened stride and spontaneous tremors (Appendix A) that started at ~5 months of age and then progressed with age. Tremors could occur in all limbs. Littermate age-matched non-tg and human wildtype DCTN1 control tg mice had no gait or posture abnormalities and were fast-walking (Appendix A). In G59S-DCTN1 mice, prominent gait abnormalities and spasticity followed, and they eventually had difficulty fully extending their hind limbs and splaying their digits and adopted a clasped limb posture when lifted by the tail (Appendix A). Affected mice developed weakness accompanied by muscle wasting in hind limbs. At 8–10 months of age, motor activity (Figure 1J) was reduced significantly in G59S-DCTN1 mice compared to age-matched non-tg mice (*p* < 0.00004) and human WT-DCTN1 mice (*p* = 0.003). Human WT-DCTN1 mice at 8–10 months of age also showed lower motor activity compared to age-matched non-tg mice (Figure 1J). Disease progression over subsequent weeks resulted in the G59S-DCTN1 mice being unable to groom as weight loss and weakness became more severe. This clinical phenotype was fully penetrant. Eventually, mice developed poverty of movement and became paralyzed (Appendix A). Mice unable to right themselves after a 10 s interval were considered at endstage of the disease and were euthanized for pathological examination. The mean lifespan of the G59S-DCTN1 mice was ~400–500 days, consistent with our original study [23]. Human wildtype DCTN1 tg mice did not develop major clinical disease.

### 3.3. Motor Neurons Degenerate in Mutant Dynactin Tg Mice

In this study, we focused mostly on the neuropathology of the spinal cord. Motor neurons were counted in cresyl-violet-stained sections (Figure 1A,B) throughout the lumbar spinal cord region in age-matched mice expressing human wildtype DCTN1 or mutant DCTN1 at endstage disease. Particular attention was paid to the matching of spinal cord sections for similar anatomical levels. Mutant mice had a significant (*p* < 0.0001, ~50% loss) reduction in the total number of lumbar motor neurons compared to wildtype-DCTN1 mice (Figure 1G). Histologically with cresyl violet staining, normal mouse spinal motor neurons had large multipolar polygonal cell bodies (40–50 µm in major diameter), enrichment of cytoplasmic Nissl substance, and a prominent nucleus with a nucleolus (Figure 1C). Some motor neurons in mutant DCTN1 mice were classified as damaged because they had cytoplasmic vacuoles and sidled small mononuclear cell satellites near the cell periphery (Figure 1D and Figure 2). Damaged motor neurons were also found in mice expressing human WT-DCTN1 (Figure 1H). The ratio of damaged motor neurons to total motor neurons was variable but slightly higher in G59S-DCTN1 mice compared to WT-DCTN1 tg mice (Figure 1I). However, motor neurons in chromatolysis with rounded cell bodies and pale attenuated cytoplasmic Nissl substance and eccentric nuclei (Figure 1E) and motor neurons that were round, pale, and mostly depleted of Nissl substance (Figure 1F) were observed only in mutant DCTN1 mice (Figure 1E,F).

### 3.4. Cytotoxic T Lymphocytes and Inflammatory Cytokines Accumulate in Dynactin Transgenic Mouse Spinal Cord

Cresyl-violet-stained sections suggested that small mononuclear cells accumulate in the spinal cords of DCNT1 mice. Thus, we stained for Fas (CD95) using a monoclonal antibody shown previously to be highly specific [39]. Small mononuclear cells typical of lymphocytes were positive for Fas in the spinal cords of non-tg, human wildtype DCNT1 tg mice and G59S-DCTN1 mice (Figure 2). Notably, in the human G59S-DCTN1 mice, small mononuclear-positive cells were in apparent apposition to motor neurons (Figure 2B) and could be seen affiliated with disintegrating motor neurons in mutant mice (Figure 2C). Some Fas-positive cells appeared as cell clusters or nests around motor neurons. Individual Fas-positive small cells in lumbar spinal cord were counted in non-tg and tg mice (Figure 2D) independent of cell clustering. Compared to non-tg mice, Fas-positive cell counts were significantly increased in human wildtype DCTN1 tg mice (*p* = 0.02) and G59S-DCTN1 mice (*p* < 0.0001). Fas-positive cell counts were higher significantly (*p* = 0.00009) in human G59S-DCTN1 mice compared to human wildtype DCTN1 tg mice. Immunohistochemistry for CD8 also revealed an accumulation of CD8-positive cells selectively in the ventral horn gray matter (Figure 2E) and white matter near the ventral root exit zones (Figure 1F) of G59S-DCNT1 mice. CD8-positive cells were not seen in the spinal cord of age-matched non-tg and WT-DCTN1 tg mice.

Because of the invasion of T cells into the spinal cord of G59S-DCTN1 mice, we profiled two different inflammatory cytokines (Appendix A). Immunoreactivity for TNFα was very low in the spinal cord of human WT-DCTN1 mice (Appendix A) but was highly enriched in the spinal cord ventral horn of G59S-DCTN1 mice (Appendix A). Early in symptomatic disease, small TNFα-positive cells accumulated in the ventral horn parenchyma in the vicinity of motor neurons (Appendix A, arrows), and as disease advanced, damaged motor neurons with vacuoles adopted a TNFα-positive phenotype as small TNFα-positive cells were seen present in the parenchyma (Appendix A). In advanced disease, severely damaged vacuolated and lytic motor neurons remained positive for TNFα (Appendix A, arrows).

Like the TNFα results, IL9 immunohistochemistry also revealed a pattern suggestive of an inflammatory cell attack and lytic process in motor neurons of G59S-DCTN1 mice. Human WT-DCTN1 spinal cord sections were essentially devoid of IL9 staining (Appendix A). In contrast, human G59S-DCTN1 mice had perivascular and parenchymal infiltration of small IL9-positive cells (Appendix A). As disease advanced, damaged motor neurons with vacuolations in G59S-DCTN1 mice showed numerous pericellular decorations of small IL9-positive cells (Appendix A).

### 3.5. Mutant Dynactin Mice Develop Axon Pathology

The chromatolytic reaction of motor neurons is generally believed to be a response to axonal injury [19,20,21,40,41]. We therefore examined 1 µm thick plastic (Epon) sections of human DCTN1 tg mouse spinal cord and sciatic nerve that favorably disclose axons and evidence of axonopathy that is reliably quantifiable (Figure 3E,F). Degenerating axonal profiles were prominent in the spinal cord anterior white matter commissure (Figure 3A), ventral root exit zones (Figure 3B), and dorsal corticospinal tract (Figure 3C) in G59S-DCTN1 mice. Occasionally, damaged axons were seen in human WT-DCTN1 tg mice (Figure 3D). In proximal sciatic nerve sections, the total number of normal axons was significantly lower (*p* < 0.001) in G59S-DCTN1 mice compared to age-matched human wildtype DCTN1 mice (Figure 3E). The number of degenerating axon profiles was significantly greater (*p* < 0.001) in G59S-DCTN1 mice compared to age-matched human wildtype DCTN1 mice (Figure 3F). To visualize axonopathy optimally in vivo with an alternative approach, we bred tg G59S dynactin mice to tg mice expressing yellow fluorescent protein (YFP) in neurons. Florid YFP-positive axonopathy was observed in spinal cord and brainstem in double-tg mice (Figure 3E).

### 3.6. Mutant Dynactin Mice Have Extensive Mitochondrial Pathology

While studying axonopathy in 1 µm thick Epon sections, we observed that mitochondria appeared to accumulate in the cell bodies of motor neurons of G59S-DCTN1 mice (Figure 4B inset). To directly identify mitochondria specifically in motor neuron cell bodies, we used immunohistochemical staining for SOD2 that was localized within the mitochondria matrix [42]. For this experiment, we examined mice at early presymptomatic stages of disease (4 months of age, *n* = 6) and at symptomatic stages of disease (8 months of age, *n* = 6). With light microscopy, mitochondria were seen as individual particles or poly-mitochondrial congregations within the cytoplasm of motor neurons (Figure 4A,B). Under 1000× oil magnification and careful *z*-axis focusing, mitochondrial profiles were counted within individual motor neurons in human wildtype and mutant DCTN1 tg mice (Figure 4F). Motor neurons in G59S-DCTN1 mice at 4 months of age had significantly more (Figure 4F, *p* = 0.001) perikaryal total mitochondria compared to human WT-DCTN1 tg mice. In contrast, at 8 months of age (Figure 4F), mutant mouse motor neurons had significantly fewer mitochondria compared to 4-month-old tg mutants (*p* < 0.001) and 8-month-old tg wildtype mice (*p* = 0.03). When motor neuron mitochondria were fractionated based on particle size, 4-month-old mutants had significantly smaller mitochondria than 4-month-old wildtype mice (Figure 4G, *p* = 0.003). At 8 months, tg mutants had fewer small mitochondria compared to 4-month tg mutants (Figure 4G, *p* = 0.04). Interestingly, as the fraction of small mitochondria dissipated in the mutant motor neurons at 8 months of age, the fraction of swollen mitochondria increased compared to the mutants at 4 months and the wildtypes at 8 months of age (Figure 4H). These dynamics in mitochondria are illustrated pictorially in the mutants with the accumulation of SOD2 immunoreactivity compared to wildtype (Figure 4A,B) and the subsequent loss of SOD2 immunoreactivity (Figure 4C). As motor neurons in the mutant mice entered the chromatolytic stage of degeneration characterized by somal rounding, nuclear eccentricity and cytoplasmic pallor, the overall mitochondrial number decreased significantly. Some individual mitochondria were prominently round and swollen (Figure 4D). Motor neurons at the endstage of their degeneration, suggested by their pale dissolution, were nearly devoid of cytoplasmic SOD2 immunoreactivity (Figure 4F).

### 3.7. Targeting Mitochondria with Small-Molecule Drugs Mdivi-1 and GNX-4728 Protects Motor Neurons in Mutant Dynactin Mice

The prominent mitochondrial changes that occur in motor neurons of mutant dynactin mice could signify a perturbation in mitophagy [43]. Or they could signify secondary bystander changes or primary drivers in the process of neurodegeneration involving aberrant axonal transport and accumulation or fission of mitochondria akin to a Trojan-horse-like pathology [34,43] associated with enhanced oxidative stress [41,44,45]. To explore the possibility of abnormal mitophagy, we stained spinal cord sections for Parkin (Appendix A). Major differences in Parkin localization were not seen in motor neurons of human WT-DCTN1 and G59S-DCTN1 tg mice (Appendix A). Some chromatolytic motor neurons in G59S-DCTN1 mice showed perikaryal margination of Parkin immunoreactivity and diffuse particulate immunoreactivity with the cytoplasm (Appendix A). The most dramatic differences in Parkin immunostaining in WT- and G59S-DCTN1 tg mouse spinal cords were the accumulation of Parkin in swollen dystrophic axons and apparent astrocytes in the ventral funiculus and ventral root exit zones (Appendix A).

To test the mitochondrial fission hypothesis in vivo, mice were treated with BrdU to label mtDNA replication (Figure 5A–C). With the BrdU pulse delivered, mtDNA replication was barely or not detectable in human wildtype DCTN1 mouse motor neurons (Figure 5A), but cell proliferation was detected in the central canal epithelia (Figure 5A inset). In contrast, cytoplasmic particulate BrdU labeling was detected in G59S-DCTN1 mouse motor neurons (Figure 5B,C).

The BrdU experiment supported the idea of enhanced mitochondrial fission in mutant dynactin mice. We therefore treated mutant dynactin mice intracerebroventricularly with the mitochondrial division/fragmentation inhibitor Mdivi-1, starting at a presymptomatic stage of disease with weekly doses over 4 weeks. Mdivi-1 attenuated the mitochondrial pathology in spinal motor neurons of mutant dynactin mice (Figure 5D–F). Importantly, mutant dynactin mice treated with Mdivi-1 compared to mutant dynactin mice treated with vehicle showed the following: (1) a significant rescue of the total number of motor neurons (Figure 5G, *p* < 0.001); (2) a significantly reduced number of surviving motor neurons with a damaged phenotype (Figure 5H, *p* < 0.04); (3) a modest difference in the ratio of damaged motor neurons to total motor neuron (Appendix A); (4) significantly (*p* = 0.00005) improved motor activity (Figure 5I); and (5) a significant extension of mean lifespan compared to vehicle-treated G59S-DCTN1 mice (Figure 5J).

Our G59S-DCTN1 mice developed a mitochondrial swelling phenotype. Because mitochondrial swelling can be driven by the mitochondrial permeability transition pore (mPTP) [28,29], we also tested the therapeutic efficacy of a small-molecule drug (GNX-4728) that inhibits the mPTP and has shown positive effects in aggressive mouse models of ALS [29]. In a small-group-sized pilot study, systemic treatment of G59S-DCTN1 mice significantly extended their lifespan (Appendix A).

### 3.8. Motor Neurons in Mutant Dynactin Have Activated Caspases

The spinal cord motor neurons in mutant dynactin mice appear to develop many features of retrograde neurodegeneration, including chromatolysis and the axon reaction, mitochondrial accumulation, and motor neuron loss. We have studied these phenotypes before in other rodent models of spinal motor neuron cell death [34,39,41]. Because of these similarities with other rodent models, we examined the mutant dynactin mice for evidence of caspase activation and cell death.

Immunohistochemistry revealed evidence for caspase-3 activation in the spinal cord neuropil and motor neuron cell bodies of mutant dynactin mice but not in age-matched human wildtype dynactin mice (Figure 6A,B). Subsets of motor neurons were conspicuously positive for cleaved caspase-3 in mutant mice (Figure 6C). The ratio of cleaved-caspase-3-positive motor neuron cell bodies to total neurons was increased significantly (*p* < 0.0001) compared to wildtype mice (Figure 6D,F). The presence of cleaved caspase-3 in morphologically identified chromatolytic motor neurons suggests that they are dying [41,42,43,44,45,46]. An in situ DNA fragmentation assay with the TUNEL method confirmed that motor neurons in mutant dynactin tg mice were undergoing cell death (Figure 6F), thus accounting for their reduced number, while TUNEL-positive motor neurons were not seen in human wildtype dynactin tg mice (Figure 6E).

When it occurs, caspase-3 activation is believed to be a relatively far-downstream event in the cell death process [47,48]. We therefore performed immunohistochemical staining for activated caspase-8 to determine if this initiator arm of the caspase signaling pathway was also activated in motor neurons of dynactin mice (Figure 7A,B). Cleaved-caspase-8-positive motor neurons in mutant dynactin mice were increased significantly compared to age-matched wildtype dynactin tg mice (Figure 7D, *p* < 0.001). Interestingly, while the cleaved-caspase-3-positive neurons were usually in the chromatolysis stage of degeneration, the cleaved-caspase-8-positive motor neurons were often multipolar (Figure 7C) suggesting caspase-8 activation at an earlier stage of motor neuron degeneration. This finding is consistent with T cell-mediated cytolysis.

### 3.9. Motor Neurons in Mutant Dynactin Tg Mice Develop Hsp90 Nitration

Considerable data has accrued inviting the concept that protein nitration can drive the mechanisms of motor neuron degeneration in human ALS and in animal and cell models of ALS [49,50,51]. More recently, specific nitration of Hsp90 has been shown to be a critical driver of motor neuron degeneration in different model systems and possibly human ALS [37]. We therefore searched for evidence of Hsp90 nitration in mutant dynactin mice. We used a highly specific antibody that detects Hsp90 when nitrated at tyrosine residue-56 (Hsp90_NY56_) [37]. Wildtype dynactin tg mice had very low immunoreactivity for nitrated Hsp90_NY56_ (Figure 8A) in their spinal cord ventral horn, consistent with their non-chromatolysis multipolar morphology, as demonstrated by Nissl staining (Figure 8B). In contrast, in lumbar spinal cord of age-matched G59S dynactin mice, many motor neurons were positive for nitrated Hsp90_NY56_ (Figure 8C,F). Immunopositivity of motor neurons was age-related and appeared emergent with the chromatolytic change (Figure 8E,F).

## 4. Discussion

*DCTN1* gene mutations cause varying clinical neurology presentations in humans. The diseases associated with *DCTN1* gene mutations include ALS, ALS-FTD, atypical Parkinson’s disease, Perry syndrome, spinal and bulbar muscular atrophy, and distal hereditary motor neuronopathy type 7B [52,53]. The association of the *DCTN1* gene with ALS was discovered in 2003 [1]. Subsequently, mutations in *DCTN1* were found to be causative in Perry syndrome [54,55] and other neurodegenerative diseases [56]. However, these *DCTN1* pathogenic mutations are thought to be rare overall in occurrence [56]. Nevertheless, the specific, therapeutically relevant mechanisms through which these gene mutations cause neurodegeneration are unknown, and, thus, there are no disease-modifying treatments or cures for patients with *DCTN1*-related gene mutations.

*DCTN1* encodes the dynactin-1 subunit protein (p150) of the dynactin complex that contains eleven different proteins [6,7]. The dynactin-1 subunit is the largest protein in the complex and critically interacts with dynein and microtubules for the intracellular retrograde transport of cargo, including mitochondria and neurotrophin-containing vesicles [18,57,58]. The dynactin-1 subunit has a cytoskeleton-associated protein glycine-rich domain that mediates the binding of the dynactin complex to microtubules [6,7]. It is required for the initiation of dynein-driven cargo transport from the distal axon by recruiting and tethering dynein to microtubules [59]. The tg mouse line used here has a missense substitution mutation at amino acid 59 that replaces a glycine with a serine residue (G59S). This specific mutation (G59S) was one of the first discovered to cause a *DCTN1*-linked slowly progressive autosomal dominant form of human ALS [1,2]. There are several other point mutations in *DCTN1* that have been identified in humans [3]. Most tg mouse modeling has focused on the G59S mutation. Tg mice harboring this mutation have been generated [23,60,61]. Importantly, the outcomes of these experiments were different, despite two studies using similar Thy1.2 promoters. A G59S-DCTN1 heterozygous knock-in mouse showed a modest age-related phenotype with neuromuscular junction abnormalities and slight motor neuron loss but no lifespan shortening [60]. In another study, G59S-DCTN1 mice developed adult-onset, slowly progressing, muscle weakness with distal axonopathy and motor neuron lipofuscin accumulation but no motor neuron loss or mouse lifespan shortening [61]. Female mice generally were more affected than male mice, perhaps because of the X-chromosome of the transgene [61]. Our original study found that Thy1-G59S-DCTN1 mice also developed adult-onset, slowly progressing, skeletal muscle weakness with ventral root axonopathy, and there was a robust motor neuron degeneration phenotype, and the mice developed fatal disease with shortened lifespan [23]. Female and male mice were affected about equally [23]. The differences in the outcomes of these studies could be due to background mouse stain differences or transgene insertion sites. Here, after several generations, our original findings are confirmed and substantially advanced regarding the details and mechanisms of the degeneration of motor neurons. We also show motor function data and video evidence that these tg mice have a clinical phenotype consistent with ALS, including progressively developing ataxia and apparent spasticity, weakness, skeletal muscle atrophy, and paralysis. We also confirm the fatal phenotype and show that mitochondria can be targeted in two ways to extend the lifespan of these mice. Arguably, the slow progression of neurologic disease, spanning for clinical onset at 5–6 months of age to progression over another 10–12 months, and the specific clinical features of the disease in G59S-DCTN1 tg mice (see Appendix A) better mirror human ALS than the more commonly used tg SOD1 mouse models [34]. Neuropathologically, G59S-DCTN1 tg mice develop lower (spinal motor neurons) and upper (dorsal corticospinal tract axonopathy) motor neuron disease.

We found that our G59S-DCTN1 mouse model of ALS developed a prominent mitochondrial phenotype in motor neurons. This phenotype was age-related and involved mitochondrial accumulation in the motor neuron perikaryon and mitochondrial dysmorphia such as apparent division or fragmentation and enlargement. The apparent replication of mtDNA detected with BrdU labeling supported the evidence for the expansion of the mitochondrial pool based on SOD2 labeling. Motor neurons in the chromatolytic stage of degeneration had mitochondria that adopted a round shape prior to the soma becoming SOD2-negative. Mitophagy did not appear to be prominent as a mechanism for mitochondrial clearance within motor neuron cell bodies because immunostaining for Parkin, which functions in mitophagy [44], was largely unremarkable. Moreover, prior electron microscopy assessments suggested no evidence for increased mitophagy in motor neuron cell bodies in these mice [23]. However, we did find some accumulation of Parkin immunoreactivity marginated at periphery of the chromatolytic motor neuron cell bodies and, more strikingly, accumulation of Parkin in swollen degenerating axons. Prominent mitochondriopathy has been seen in motor neurons of other tg mouse models of ALS, notably SOD1 models [34,62,63,64]. This study is the first to show aberrant mitochondrial localization and morphology in motor neurons of G59S-DCTN1 tg mice and to use this information on mitochondria as a platform for therapeutics in these mice. The participation of mutant DCTN1 in abnormal retrograde transport of mitochondria has been articulated before [65], and this involvement is made more fascinating by the discovery of trafficking kinesin proteins [66]. Other work on zebrafish has shown elegant relationships between altered functions of the dynein–dynactin complex and mitochondrial trafficking and localization in axons [57]. Our specific finding of the mitochondrial rounding morphology in G59S-DCTN1 motor neurons has been seen in mutant SOD1 models [64,67]. It is possible that the mitochondrial shape pathology, the increased mtDNA replication, and the Fas-positive T cell invasion shown here in G59S-DCTN1 mouse spinal cord are coupled to mtDNA leakage in motor neurons [68,69]. Mitochondria can release mtDNA as a potent damage-associated molecular pattern [68,69] that can stimulate cyclic GMP-AMP synthase-stimulator of interferon gamma signaling-dependent T-cell-mediated cytolysis [70,71]. These pathological processes, including mtDNA leakage, in G59S-DCTN1 mice, and possibly in human DCTN1-related neurodegenerative diseases, could emanate and converge at the mitochondrial permeability transition pore (mPTP), as described for other clinical settings and in other experimental model systems [72]. The mPTP has been shown to be a relevant target for therapy in mutant SOD1 tg mouse models of ALS [29]. Our pilot study here with the mPTP inhibitor GNX-4728 suggests that the mPTP might also be a direction for therapeutic discovery in mutant DCTN1 models because it modestly extended lifespan. This pilot data needs to have a workup of the neuropathology in G59S-DCTN1 mice treated with GNX-4728 to determine if mitochondrial swelling and other forms of mitochondriopathy were lessened.

Most of our ideas on the mitochondrial-related mechanisms of disease in mutant DCTN1 mice focus directly on mitochondria. However, mitochondrial pathology and disease mechanisms can be engaged in the nucleus. We do see nuclear phenotypes in the G59S-DCTN1 mice, including the nuclear eccentricity of the axon reaction and the nuclear accumulation of nitrated-Hsp90. Additional studies also need to address the possible pre-mitochondrial nucleus-to-mitochondria signaling pathways that could be perturbed in G59S-DCTN1 mice.

A robust neuropathological signature of the neurodegeneration in G59S-DCTN1 mice was the accumulation of small mononuclear inflammatory-like cells near degenerating motor neurons. This finding kindled the idea of a cytotoxic T-cell-mediated cell death process in these mice. The striking accumulation of Fas-positive T cells in the spinal cord near motor neurons supports the idea. Though Fas (CD95) is not a specific marker for cytotoxic T cells, many subsets of T cells have cytotoxic potential [73,74]. CD8 is recognized as more specific for CNS infiltrating cytotoxic T cells [75]. Invasion of CD8-positive T cells was very specifically detected in G59S-DCTN1 mouse spinal cord, with no or negligible CD8-positive cells in human WT-DCTN1 tg mice. The magnitude of the CD8-positive T cell invasion was much less compared to the invasion of CD95 cells in G59S-DCTN1 mice. However, the localization of CD8 immunoreactivity was very focal and confined largely to the ventral horn and nearby white matter where degeneration was seen. In situ cytokine profiling with antibodies to TNFα and IL9 revealed robust accumulation of these cytokines in the ventral horn and nearby white matter of G59S-DCTN1 mice. Small TNFα- and IL9-positive cells could be found decorating motor neurons with strikingly damaged and lytic morphology. TNFα has been implicated in the neuropathology of human ALS and SOD1 mouse models of ALS, but its role in ALS mechanisms of disease is unclear [76]. CD4-positive cells are the main source of IL9 [77]. IL9-producing T cells have robust cytotoxic potential in tumors [78] and have been implicated in injury to white matter in the brain [79]. Future studies are needed to demonstrate that the IL9-positive small cells encrusting motor neurons in G59S-DCTN1 mice are CD4-positive T cells. This is important because both CD8- and CD4-positive T cells accumulate in vulnerable brain and spinal cord regions in human ALS [80].

We found that the degenerating motor neurons in the G59S-DCTN1 mice were positive for cleaved caspase-8 and cleaved caspase-3, and they became TUNEL-positive. This has not been shown before in other studies of mutant DCTN1 mouse models. We have rigorously characterized these antibodies for their specific detection of cleaved subunits in the absence of proenzyme detection by Western blotting and mass spectroscopy [34,39]. Cleaved caspase positivity, notably caspase-8, would be consistent with T-cell-mediated death of target cells [81]. This idea is supported further by (1) the invasion of CD8-positive cells into the spinal cord; (2) the propinquity of Fas-positive T cells to the degenerating motor neurons; and (3) the accumulation of TNFα and IL9 immunoreactivities in the spinal cord ventral horn and in small cells decorating damaged motor neurons. The cleaved caspase-3 findings were also noteworthy because of the discrete localization of cytoplasmic aggregations within chromatolytic motor neurons in G59S-DCTN1 mice. This observation mirrors our findings in chromatolytic motor neurons in human ALS [82]. These discrete caspase-positive structures might represent accretions of assembled heptameric 1-megaDalton apoptosomes [83] that have microscopic resolution [84]. Our in vivo findings are consistent with cell culture work showing that G59S-DCTN1 causes activation of the caspase-dependent apoptotic pathways in HeLa and SH-SYSY cells [85]. However, our results are fascinating because the motor neuron death was decidedly not morphological apoptosis, despite the activation of two different caspases. Other forms of cell death along the cell death continuum need to be examined in the mice, though likely possibilities include necroptosis and autophagy [23,47].

We discovered that mitochondria are feasible therapeutic targets for protecting motor neurons and extending the lifespan of the G59S-DCTN1 mouse model of ALS. We used two fundamentally different drugs (Mdivi-1 and GNX-4728) that target two different aspects of mitochondrial biology. To corroborate our BrdU and SOD2 findings suggesting an expansion of the mitochondrial pool in motor neurons in G59S-DCTN1 mice and to explore the mitochondrial changes as disease mechanisms, we tested the small molecule Mdivi-1 in mice as a therapeutic. Mdivi-1 is an inhibitor of dynamin-related protein-1 (Drp1) [86]. Drp1 is required for mitochondria fission [87]. We showed that Mdivi-1 protected motor neurons from loss in spinal cord and extended the lifespan of G59S-DCTN1 tg mice. Pharmaceutically targeting Drp1 with peptide inhibitor has been therapeutically successful in cell culture models of ALS [88]. Inhibition on Drp1 with Mdivi-1 specifically has been shown to reduce functional deficit in a retrograde toxin model of motor neuron degeneration in mouse [89] and provide neuroprotection in a rat model of Parkinson’s disease [90]. Here, we delivered Mdivi-1 ICV starting at presymptomatic stages of disease. Drug administration after the clinical diagnosis would be more clinically relevant. A future mouse experiment could address this issue. More translationally relevant drug delivery approaches need to be developed instead of using ICV delivery. Different formulations of Mdivi-1, and similar drugs, need to be developed. Mdivi-1 as a therapeutic has important caveats. In addition to inhibiting Drp1, the drug has effects on mitochondrial oxidative metabolism and reactive oxygen species by reversibly inhibiting complex I, independent of its effect on Drp1, and it has effects on intracellular Ca^2+^ dynamics [86]. Some of these non-Drp1-related drug effects of Mdivi-1 could be responsible for the residual damage seen in surviving motor neurons. Thus, the mechanisms underlying the therapeutic effects of Mdivi-1 in G59S-DCTN1 tg mice need more study.

We found that spinal cord motor neurons in G59S-DCTN1 tg mice showed positivity for Hsp90 nitrated at Y56. This phenotype was age-related and found mostly in neurons at the chromatolytic stage of their degeneration and was enriched in the cytoplasm. Motor neurons with positivity for Hsp90_NY56_ have been found previously in human ALS and in a mutant SOD1 mouse model of ALS [37]. Protein nitration, identified by 3-nitrotyrosine antibodies in general and antibodies to specifically nitrated target proteins, is a footprint for the presence of peroxynitrite [49]. Peroxynitrite formation, derived from the reaction of nitric oxide with superoxide [49], has been implicated in the mechanisms of motor neuron apoptosis in cell culture [50,91]. The motor neuron genome appears to be particularly vulnerable to damage by peroxynitrite [92]. Immunoreactivity for 3-nitrotyrosine has been found to be elevated in motor neurons in human ALS postmortem spinal cord [93,94,95] and in motor neurons in a mutant SOD1 mouse model of ALS [96]. The finding that motor neurons in mutant dynactin tg accumulate immunoreactivity for Hsp90_NY56_ is novel and particularly interesting because many were morphologically chromatolytic. This process is displayed consistently in motor neurons when their axons are damaged, and their cell bodies undergo fundamental alterations in metabolism [40,41,97,98]. Chromatolysis is regularly seen in human ALS motor neurons [99,100]. Motor neurons undergoing chromatolysis can fully degenerate and die or they can repair and recover [19,20,40], but the molecular controls for commitment to death or to survival are not understood [98]. Our mutant dynactin tg mouse model had robust motor neuron chromatolysis and cell death, in which about 50–60% of lumbar spinal cord motor neurons were eliminated. During this process, many chromatolytic neurons became positive for nitrated Hsp90. It thus is possible that the motor neuron commitment to death might involve nitrated Hsp90. This involvement could be related to the capacity of Hsp90_NY56_ to stimulate glycolysis and to activate Ca^2+^ channels [101], possibly assisting in lactate-driven intracellular acidification, mitochondrial swelling, and cytolysis [102].

Based on the SOD2, mitochondrial swelling, and Hsp90_NY56_ findings, an mPTP inhibitor drug (GNX-4728) was tested in a small pilot study for therapeutic potential in G59S-DCTN1 tg mice. Prior work has shown GNX-4728 to be a very robust therapeutic in mutant SOD1 tg mice [29]. We found that GNX-4728 also extended lifespan in DCTN1 mice. This pilot study was very limited in scope, but it might suggest that the mPTP is a convergence point for the varying pathologic mechanisms for motor neuron degeneration that we describe here in G59S-DCTN1 tg mice.

## 5. Conclusions

We characterized tg mice expressing either human wildtype or mutant (G59S) DCTN1. Mice with G59S-DCTN1 develop profound age-related disease consistent with ALS. Mice with human wildtype DCTN1 did not develop clinical disease or major neuropathology. The disease phenotypes of G59S-DCTN1 tg mice included diminished motor activity, eventual paralysis, shortened lifespan, and motor neuron degeneration and loss. This motor neuron degeneration evolved with profound axonopathy and retrograde neurodegenerative phenotypes, mitochondrial pathology, caspase activation, inflammatory pathology, and T-cell-like motor neuron-targeted cytolytic neurodegeneration. Two different mitochondrial acting drugs were therapeutic in these mice.

## Figures and Tables

**Figure 1 biomolecules-15-01637-f001:**
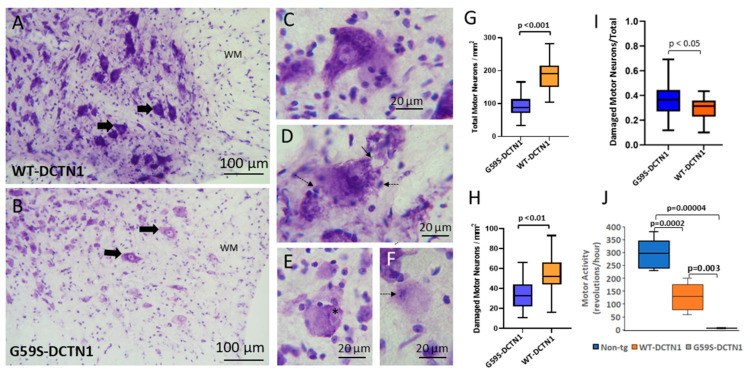
Cresyl-violet-stained lumbar spinal cord sections detailing motor neuron degeneration in human mutant dynactin G59S tg mice. (**A**) Human wildtype DCTN1 mouse spinal cord cross-section with prominent motor neurons (solid broad arrows) within the ventral horn. The motor neurons are numerous, clustered, large (roughly 25 µm in diameter), multipolar, and generally enriched in cresyl violet staining. The white matter (wm) lateral funiculus is identified. (**B**) Human mutant DCTN1 mouse spinal cord cross-section revealing apparently fewer motor neurons (solid broad arrows) with weaker Nissl staining within the ventral horn. The ventral horn appears depleted of motor neurons. Faint cresyl violet staining is indicative of motor neuron degeneration. Some motor neurons are evidently in chromatolysis with pale cytoplasm, rounding shape, and eccentrically placed nucleus (open arrow). (**C**) Normal-appearing motor neurons in WT-DCTN1 mouse spinal cord. They are defined as normal because of their large size (~25 µm in minor diameter), no attrition, multipolar shape, non-vacuolated cytoplasm with high Nissl enrichment, and uncondensed open nucleus. (**D**) Damaged motor neurons in G59S-DCTN1 mice had cytoplasmic vacuoles (solid thin arrows), as evidenced by disruption of the cytoplasmic Nissl staining. The damaged motor neurons often appeared to be contacted by small, dark staining, non-neuronal mononuclear round cells (dashed arrows). (**E**) Motor neurons in G59S-DCTN1 mouse spinal cords degenerated in a process involving chromatolysis characterized by rarefaction of the normal Nissl-rich cytoplasm starting at the cell periphery, rounded somal contour, and a nucleus (*) that is eccentrically placed. Chromatolytic neurons often had small mononuclear cell satellites. (**F**) Motor neurons in mutant DCTN1 mouse spinal cords appeared to advance to a stage of degeneration whereby the Nissl substances became nearly inconspicuous throughout the pallid cytoplasm, and the nucleus became hyperchromic throughout the nucleoplasmic matrix (dashed arrow). (**G**) Box plot showing means (with IQR and 5–95 percentile whiskers, *n* = 6 mice/group) of lumbar spinal cord (L3-5) motor neuron counts in wildtype (control) and mutant DCTN1 tg mice at endstage disease. Mutant mice had significantly (*p* < 0.001) fewer total motor neurons/mm^2^ in the ventral horn than wildtype mice. (**H**) Graph of the number of motor neurons showing morphological damage (see panel (**D**)) in WT- and G59S-DCTN1 tg mice. Human WT-DCTN1 tg mice had significantly more (*p* < 0.01) motor neurons/mm^2^ in the ventral horn that had some form of vacuolar damage compared to mutant mice. (**I**) Graph of the ratio of the damaged motor neurons to total motor neurons. (**J**). Running wheel motor activity in age-matched non-tg, WT-DCTN1 mice and G59S-DCTN1 mice assessed at 8–10 months of age (*n* = 6/group).

**Figure 2 biomolecules-15-01637-f002:**
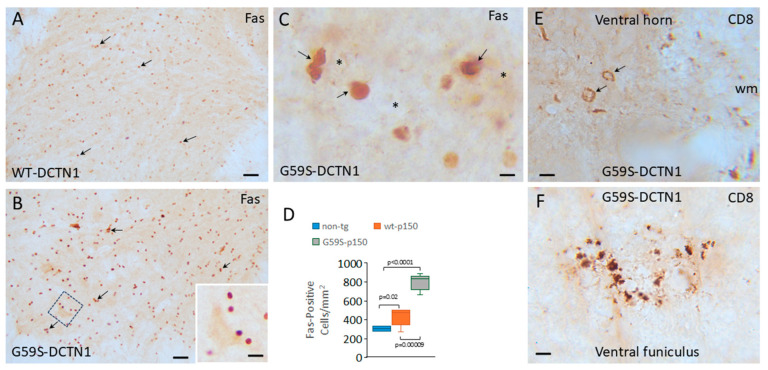
T cells invade the spinal cord parenchyma of mutant-DCTN1 tg mice. (**A**) Localization of Fas-positive cells (thin arrows) in ventral horn of wildtype DCTN1 transgenic mouse. Scale bar (same for (**B**)) = 40 µm. (**B**) Fas-positive cells (thin arrows) in the ventral horn of mutant DCTN1 tg mice appear more numerous than in wildtype tg mice. Inset is dashed square showing Fas-positive cells in the vicinity of a motor neuron. Inset scale bar = 12 µm. (**C**) Fas-positive cells (thin arrows) near degenerating motor neurons (asterisk) in mutant-DCTN1 mice. Scale bar = 7 µm. (**D**) Box plot showing the mean Fas-positive cell number (with IQR and 5–95 percentile whiskers, *n* = 6/group) in the ventral horn of age-matched non-tg, human wildtype DCTN1 (p150) and human mutant DCTN1 (p150) at 8 months of age. (**E**) Infiltrated CD8-positive cells (arrows) in the ventral horn of mutant DCTN1 mice. CD8 immunoreactivity is also present in the surrounding white matter (wm). Scale bar = 12 µm. (**F**) Focal accumulations of CD8 immunoreactivity were seen in the ventral root exit zones in the ventral funiculus of mutant DCTN1 mice. Scale bar = 15 µm.

**Figure 3 biomolecules-15-01637-f003:**
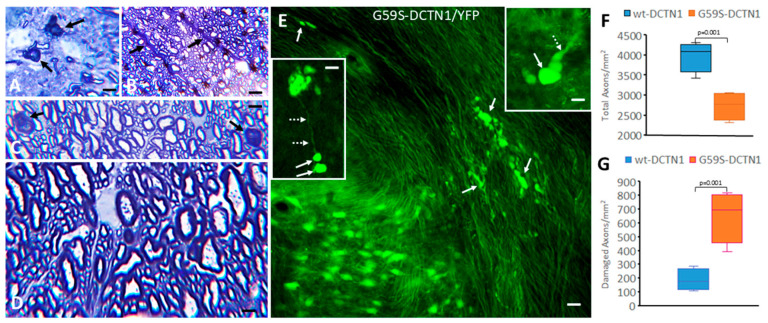
Mutant DCTN1 mice develop severe axonopathy. Epon sections (left panels) reveal degenerating axons (black arrows) in the spinal cord anterior white matter commissure (**A**), ventral root exit zone (**B**), and dorsal corticospinal tract (**C**) of mutant tg mice. Scale bars = 20 (**A**), 30 (**B**), and 10 (**C**) µm. (**D**) Fewer damaged axons were seen in age-matched WT-DCTN1 mice. Scale bar = 5 µm. (**E**) Human mutant DCTN1/yellow fluorescent protein (YFP) double-tg mice show severe axonopathy in white matter of upper cervical spinal cord (white arrows). The inset (upper right) shows a high magnification of a YFP-positive end-bulb swelling (solid arrow) of a degenerating axon (dashed arrow). The inset (left) shows a high magnification of YFP-positive axonal swellings (solid arrows) on a thin-caliber axon (dashed arrows). Scale bars = 10 µm (insets, 5 µm). Box plots show mean total (**F**) and damaged (**G**) axon number (with IQR and 5–95 percentile whiskers) in the sciatic nerve of age-matched human wildtype DCTN1 and mutant DCTN1 mice at 8 months of age (*n* = 6 mice/group).

**Figure 4 biomolecules-15-01637-f004:**
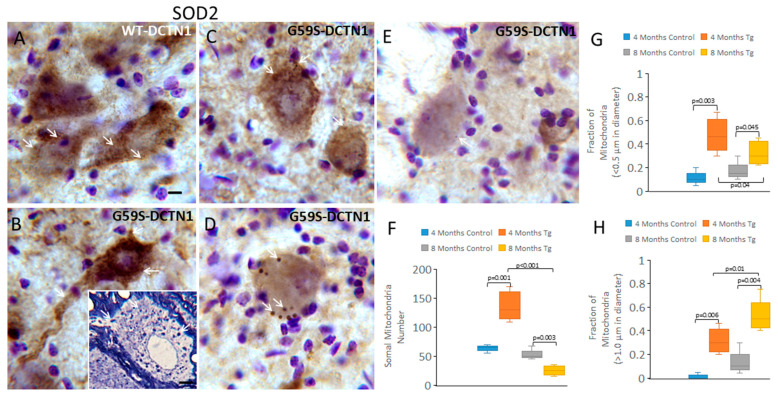
Mitochondrial abnormalities manifest in mutant DCTN1 mouse motor neurons early in disease progression and evolve with disease, as identified by SOD2 immunoreactivity. (**A**) In wildtype-DCTN1 mice, SOD2-positive mitochondria (brown) in motor neurons (shown by the purple cresyl violet counterstaining) appear as very fine particles (thin white arrows) distributed diffusely throughout the cytoplasm. Scale bar = 8 µm (same for panels (**B**–**E**)). (**B**) Mutant DCTN1 mouse motor neurons at 4 months of age are enriched with SOD2 immunoreactivity that fills the cytoplasm (thin white arrow) and proximal dendrites. Inset shows a motor neuron in a plastic section confirming the accumulation of perikaryal mitochondria (white arrows). Scale bar = 2 µm. (**C**) Mutant DCTN1 mouse motor neurons at 8 months of age show peri-plasmalemma loss of SOD2 immunoreactivity and perinuclear accumulation of apparent large aggregations of swollen mitochondria (white arrows). (**D**,**E**) Motor neurons in the chromatolytic stages of degeneration with many small mononuclear T cell satellites have scant SOD2-positive mitochondria in the cytoplasm; the residual mitochondria are swollen and round (white arrows). (**F**–**H**) Box plots show mean total mitochondria (**F**), small mitochondria (**G**), and swollen mitochondria (**H**), with IQR and 5–95 percentile whiskers, in spinal cord motor neurons of age-matched wildtype DCTN1 (control) and mutant DCTN1 mice at 4 and 8 months of age (*n* = 6 mice/group).

**Figure 5 biomolecules-15-01637-f005:**
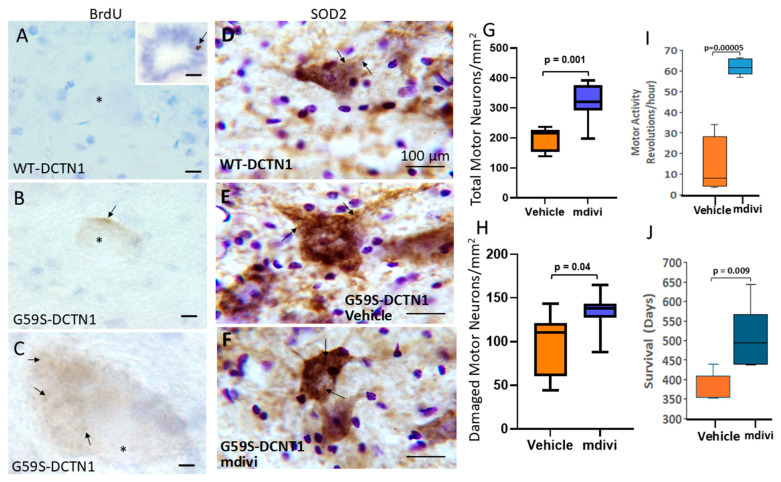
Mitochondrial division, occurring in G59S-DCTN1 mouse motor neurons, is a therapeutic target for rescuing motor neurons and extending survival. (**A**–**C**) Mitochondrial DNA (mtDNA) replication detection with BrdU. Motor neuron nuclei (*); BrdU immunoreactivity (arrows). With the BrdU pulse protocol used, mtDNA replication was below the level of detection in wildtype DCTN1 motor neurons (**A**), but positive control labeling was seen in epithelial cells lining the central canal ((**A**), inset, arrow), and labeling (arrow) was detected in mutant-DCTN1 mouse motor neurons (**B**), brown). Cytoplasmic particle localization (arrows) is shown at higher magnification (**C**). The nucleus of motor neurons was unlabeled, consistent with their postmitotic G0 status. Scale bars = 10 µm (**A**), same for (**B**), 5 µm (**C**). (**D**–**F**) SOD2 immunoreactivity shows the localization of mitochondria (arrows) in WT-DCNT1 motor neurons (**D**), the accumulation of mitochondria in mutant-DCTN1 motor neurons (**E**), and the partial mitigation of mitochondrial accumulation in motor neurons of G59S-DCTN1 mice treated with Mdivi-1 (**F**). (**G**) Box plot showing mean (with IQR and 5–95 percentile whiskers) lumbar spinal cord (L3-5) motor neuron counts in mutant DCTN1 tg mice with vehicle or Mdivi-1 treatments. (**H**) Box plot showing mean number of damaged motor neurons (with IQR and 5–95 percentile whiskers) in lumbar spinal cord (L3-5) of mutant DCTN1 tg mice treated with vehicle or Mdivi-1. (**I**) Plot of mean survival times of mutant DCTN1 tg mice treated with vehicle or Mdivi-1. (**J**) Mdivi-1 significantly extended survival of G59S-DCTN1 mice compared to vehicle-treated mutant mice.

**Figure 6 biomolecules-15-01637-f006:**
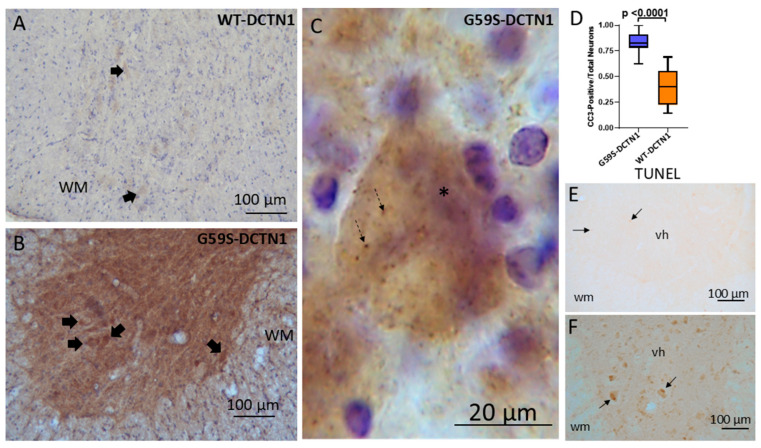
Caspase-3 activation and cell death in spinal cord motor neurons of mutant DCTN1 mice. (**A**) Human wildtype DCTN1 mouse spinal cord section immunostained for cleaved caspase-3 (with DAB detection, brown) and counterstained with cresyl violet (CV) showing very low immunoreactivity in the ventral horn neuropil and nearby white matter (WM). A few faintly labeled neurons can be discerned (solid arrows). (**B**) Human mutant DCTN1 mouse spinal cord section immunostained for cleaved caspase-3 and counterstained with CV has strong immunoreactive motor neurons (solid arrows), neuropil, and white matter (wm). (**C**) Motor neuron in mutant DCTN1 mouse spinal cord with a chromatolytic morphology, featuring an eccentrically placed nucleus (*), shows enrichment of cleaved caspase-3 immunoreactivity, some seen as cytoplasmic aggregations (dashed arrows). (**D**) Graph of the proportion of motor neurons with cleaved caspase-3 (CC3) positivity in wildtype and mutant DNTC1 (p150) tg mice. Mutant p150 dynactin tg mice showed a significantly greater (*p* < 0.0001) proportion of CC3 immunostaining in the ventral horn compared to wildtype mice. (**E**,**F**) TUNEL for DNA fragmentation shows that motor neurons in ventral horn (vh) of wildtype DCTN1 mice (**E**) are negative, but occasional small perivascular cells are positive (arrows), while subsets of motor neurons in mutant mice are positive (**F**), arrows).

**Figure 7 biomolecules-15-01637-f007:**
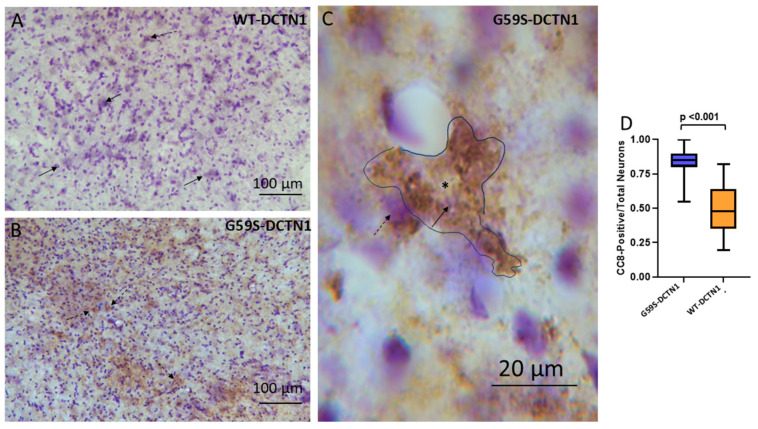
Motor neurons in human mutant DCTN1 tg mouse spinal cord are cleaved caspase-8 (CC8)-positive. (**A**) Human WT-DCTN1 mouse spinal cord cross-section showing overall weak immunostaining for ventral horn, as detected by the immunoperoxidase method with DAB as a chromogen and cresyl violet counterstaining. Motor neurons (solid arrow) and the surrounding ventral horn neuropil had minimal CC8 immunoreactivity. An occasional motor neuron had faint labeling (dashed arrows). (**B**) Human mutant p150 mouse spinal cord cross-section had a discontinuous (patchy) staining in the neuropil ventral horn, with motor neuron cell bodies showing positivity. Motor neurons (dashed arrows) and the surrounding ventral horn neuropil had higher CC8 immunoreactivity than the overall section. (**C**) Motor neuron (black outlined) in mutant G59S-DCTN1 mouse spinal cord showing accumulation of CC8 immunoreactivity in a pre-chromatolytic stage. Pre-chromatolysis is characterized by an eccentrically placed nucleus (*) and retention of a multipolar neuronal shape. The arrow points to the nucleolus in the nucleus. Apparent non-neuronal mononuclear glial cells (dashed arrow) appear to be in contact with degenerating motor neurons. (**D**) Graph of the proportion of CC8-positive motor neurons in wildtype tg (control) and mutant DCTN1 tg mice. Mutant p150 dynactin tg mice had a significantly greater (*p* < 0.001) proportion of CC8-positive motor neuros in the ventral horn compared to wildtype mice.

**Figure 8 biomolecules-15-01637-f008:**
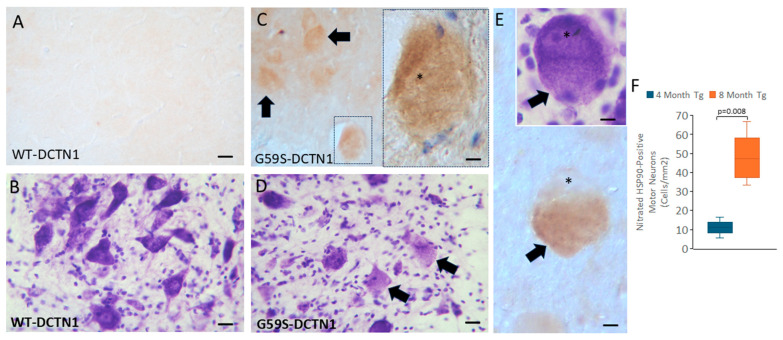
Spinal cord motor neurons in mice expressing human G59S-DCTN1 adopt a Hsp90 nitration at the Y56 phenotype. (**A**,**B**) Motor neurons in wildtype-DCTN1 (p150) mice are rarely or faintly positive for nitrated Hsp90 (**A**) and do not engage chromatolysis, as shown by cresyl violet staining (**B**). Scale bar (in (**A**)) = 20 µm (same for (**B**–**D**)). (**C**) Motor neurons in mutant G59S-DCTN1 (p150) tg mice adopt a robust nitrated Hsp90 phenotype (arrows) early in their chromatolyic phase (hatched box is shown at higher magnification on the right, illustrating a motor neuron, * identifies the nucleus in early chromatolysis prior to becoming round, scale bar = 10 µm). (**D**) Near adjacent section stained with cresyl violet verifies the chromatolytic phenotype of motor neurons (arrows). (**E**) A fully chromatolytic motor neuron (arrow) that shows robust positivity for nitrated Hsp90 in the cytoplasm while the nucleus (*) is negative. Inset (upper right) shows a similar chromatolytic independent motor neuron as seen with cresyl violet staining. Scale bar = 10 µm, inset 8 µm. (**F**) Box plot showing the mean (with IQR and 5–95 percentile whiskers) lumbar spinal cord (L3-5) motor neuron counts in mutant DCTN1 tg mice at 4 and 8 months of age that are positive for nitrated Hsp90 (*n* = 6 mice/group).

## Data Availability

The original contributions presented in this study are included in the article/Appendix A. Further inquiries can be directed to the corresponding author.

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
