# Peer review of "Human Mutant Dynactin Subunit 1 Causes Profound Motor Neuron Disease Consistent with Possible Mechanisms Involving Axonopathy, Mitochondriopathy, Protein Nitration, and T-Cell-Mediated Cytolysis"

_biomolecules, 2025, doi:10.3390/biom15121637_

Round 1

Reviewer 1 Report

Comments and Suggestions for Authors

Reviewer Comments – Biomolecules Manuscript

Title: Human Mutant Dynactin Subunit 1 Causes Profound Motor Neuron Disease Consistent with Possible Mechanisms Involving Axonopathy, Mitochondriopathy, Protein Nitration, and T Cell-Mediated Cytolysis

General Comments

This manuscript by Xie et al. presents a well-organized and comprehensive study on the pathogenic consequences of the DCTN1 (G59S) mutation in transgenic mice. The authors convincingly show that neuronal expression of mutant dynactin subunit 1 results in a progressive motor neuron disease phenotype resembling ALS, characterized by axonopathy, mitochondrial alterations, caspase activation, protein nitration, and T cell-mediated cytolysis. The data are well presented, and the inclusion of Mdivi-1 treatment adds translational relevance. The manuscript maintains a clear flow and provides valuable insights into the mechanistic basis of DCTN1-linked neurodegeneration, while identifying mitochondrial division as a potential therapeutic target.

Specific Comments

  1. In Figures 4 and 5, it would be useful to specify how “small” and “swollen” mitochondria were defined during quantification (e.g., size cut-off or morphological parameters).
  2. While t-tests are used throughout, the data involve multiple group comparisons. A short justification for this choice, or the use of ANOVA with post-hoc tests, would improve the statistical rigor.
  3. The discussion could better address known pharmacological limitations and off-target actions of Mdivi-1 to give a more balanced view of its neuroprotective effects.
  4. The discussion treats mitochondrial DNA leakage, oxidative stress, and T cell–mediated cytolysis as parallel processes. It would strengthen the manuscript to show how these mechanisms might converge and contribute collectively to neuronal loss.
  5. The reference list relies heavily on early-2000s literature, with limited inclusion of recent work (post-2019) on dynactin–dynein biology, mitochondrial dynamics, and neuroinflammation. Including more recent findings would improve the paper’s context and demonstrate engagement with current research.
  6. Recent studies indicate that mitochondrial performance differs among tissues and is shaped by the tissue-specific expression of nuclear-encoded mitochondrial genes regulating oxidative metabolism, amino acid use, and overall energy balance (for example, https://doi.org/10.1007/s00726-025-03447-4). Such nuclear–mitochondrial coordination determines how each tissue, including neurons, responds to metabolic stress or organelle injury. In DCTN1-associated neurodegeneration, this differential regulation may provide an upstream explanation for the selective vulnerability of motor neurons. The degeneration described in this study—marked by altered mitochondrial trafficking, fragmentation, and energy loss—can therefore be seen as the downstream outcome of disrupted communication between nuclear regulation of mitochondrial metabolism and cytoskeletal mechanisms that sustain organelle health. A brief note in the discussion on this aspect would strengthen the manuscript.
  7.  A light language edit would improve readability, as a few sentences are quite long and can be made more concise. I made small revisions in two areas as examples:
    - Line 99–102: “To visualize degenerating axons, wildtype- and G59S-DCTN1 transgenic mice were crossed with Thy1-YFP mice, which express yellow fluorescent protein selectively in neurons and their axons.”
    - Line 123: “Mice were euthanized by carbon dioxide inhalation and immediately perfused transcardially with ice-cold phosphate-buffered saline (PBS, pH 7.4), followed by 4% paraformaldehyde in PBS.”
  8. Figure quality – A higher-resolution version of Figure 3 or an additional inset focusing on axonal degeneration would improve the visibility of YFP-labeled structures.

Overall Assessment

Overall, this is a solid and informative manuscript. The experimental design is sound, the results are convincing, and the figures are well constructed. Addressing the above minor issues and clarifying a few methodological details would make the paper even stronger. Once revised, the work will represent a meaningful contribution to understanding DCTN1-associated neurodegeneration and mitochondrial biology.

Comments on the Quality of English Language

A light language edit would improve readability, as a few sentences are quite long and can be made more concise.

Author Response

Reviewer 1

Specific comments

  1. Specify mitochondrial size parameters.

Response. Page 5, lines 209-210. We have added text to specific cutoffs.

  1. Statistical analysis.

Response. Page 5, lines 220-228. We have redone the statistical analyses using ANOVA and post-hoc testing. Some of the p values have changed so corresponding edits have been made to several figures.

  1. In Discussion, better address the limitations and off-targets of mdivi-1.

Response. Page 18, lines 725-729. We have added content on the limitations and off-targets of Mdivi-1.

  1. It would strengthen to paper to show how mechanisms of injury might converge.

Response. Page 13, lines 494-499, Supplementary Figure 5, Discussion page 17. We have added new data to address this comment. The new data focuses on the mitochondrial permeability transition pore. We had done previously a therapeutic study on a small group of mice that were treated with the mitochondrial permeability pore inhibitor GNX-4728. The new data are presented in Supplementary Figure 5.

  1. More updated references.

Responses. We have diligently updated the literature on DCTN1.

  1. Comment on nucleus-mitochondrial signaling.

Response. We have noted the nucleus-mitochondrial signaling idea to the discussion (page 17, lines 673-679).

  1. Light language editing.

Response. We have edited the text for language.

  1. Figure 3 resolution.

Response. We have reworked completely Figure 3 and added a higher resolution inset of the axonal swelling.

Reviewer 2 Report

Comments and Suggestions for Authors

The authors investigated how the ALS-linked mutant DCTN1 (G59S) induces motor neuron degeneration, including axonopathy, mitochondriopathy, intracellular nitrative stress, and caspase activation. They further proposed that abnormal mitochondrial division may serve as a potential therapeutic target for DCTN1-related neurodegenerative diseases. The study is of significant scientific value; however, several issues and suggestions should be addressed as follows:

  1. According to the Instructions for Authors of biomolecules, the abstract should not exceed 200 words. The current abstract is too lengthy and should be shortened.
  2. The authors should clarify why neuron-specific expression of mutant DCTN1 can serve as a pathological model for ALS, and provide theoretical or literature support for this rationale.
  3. The Introduction should more clearly define the research objectives, explain the rationale for conducting this study, and highlight the novelty of the work.
  4. When assessing the motor behaviors of G59S-DCTN1 mice, why were animal behavioral tests or clinical scores not applied to quantitatively evaluate motor function?
  5. Are there any results demonstrating whether intracerebroventricular injection of Mdivi-1 alters motor performance in mice? Were behavioral tests conducted to assess this effect?
  6. Mdivi-1 was administered at the presymptomatic stage of disease. Would injection at the symptomatic stage also have therapeutic effects?
  7. Do mutations at other DCTN1 sites also cause motor neuron damage? Do mice carrying other DCTN1 mutations show similar phenotypes such as axonal degeneration or mitochondrial abnormalities?
  8. The authors report an increase in Fas (CD95)-positive cells and infer T cell–mediated cytolysis. However, Fas (CD95) is not a T cell–specific marker. CD4⁺/CD8⁺ T cell subtypes and cytokines such as TNF-α and IFN-γ should be analyzed to strengthen this conclusion.
  9. Many quantitative figures lack x-axis labels.
  10. Some staining images are missing scale bars.

Author Response

Reviewer 2

  1. Abstract is too long.

Response. We have edited the abstract to 200 words.

  1. Clarify design using neuron-specific expression.

Response. We have made this clarification in the Introduction (2nd paragraph). Notably, we wanted to isolate the cell-autonomous pathology in neurons.

  1. The Introduction could be more clearly defined.

Response. We edited the Introduction. The second paragraph better articulated the purpose of the study and are noteworthy findings.

  1. Motor function data.

Response. We have added motor function data for the mouse model. These new data are included in Figure 1 and Figure 5.

  1. Does mdivi-1 alter motor performance in mice?

Response. Yes. These new data are integrated into Figure 5.

  1. Would injection of Mdivi-1 at symptomatic stages also have worked?

Response. We address this in the Discussion section on limitations and caveats of Mdivi-1 experiments.

  1. Do mutations in other DCTN1 sites cause motor neurons damage?

Response. Other mutated sites in DCNT1 have been identified in human. However, most work has focused on the G59S mutation. We now mention this in the Discussion.

  1. Additional staining for cytotoxic T cells and cytokines is needed.

Response. We have done a lot of new staining to address this comment. This turns out to be exciting data. New immunohistochemical data is shown for CD8 (Figure 2) as well as TNFα and IL9 (as an alternative for CD4). The new cytokine data is shown in Supplementary Figure 2.

  1. X-axes.

Response. We added x-axes to the graphs. The legends identify the groups.

  1. Missing scale bars.

Response. We have added scale bars to the panels.

Reviewer 3 Report

Comments and Suggestions for Authors

Xie and colleagues provide a manuscript on a motor neuron disease mouse model by the induction of a human mutant dynactin subunit 1. Histology was used to determined motor neuron death/ apoptosis. Further studies suggested the human mutant dynactin-1 causes mitochondria death, which was prevented with mdivi-1 treatment in vivo.

There are several major concerns with the manuscript:

1) This is not the first study of a mutant dynactin model of MND in a mouse. Lai et al., (2007) demonstrated this (doi: 10.1523/NEUROSCI.4226-07.2007). Also, the senior author have published their model in 2008 in a previous article Laird et al., https://doi.org/10.1523/JNEUROSCI.4231-07.2008. Also, unclear how novel is the model when the same group has published the following article in 2008 (https://doi.org/10.1523/JNEUROSCI.4231-07.2008).

2) Line 113. surprised animals survived after giving 2 ul of fluid containing DMSO into the brain every other day for 4 weeks. This is a very large volume to be injected as a bolus into the brain. What was the well being of the mice, esp. the mutant mice after injection? was animal briefly anesthetized when injected? How was back flow prevented? Reference 20 was suggested to give the information required but had insufficient information on this part, so need more detail.

3) Insufficient to provide western blot of dynactin only in suppl. figure. Need to demonstrate the human mutant dynactin is present in the host or not. This provide an important understanding of the pathophysiology. If incorporated, then suggest it has affected the dynactin process (loss- and/or gain of function. If not incorporated, then is gain of function. Also, how was WT dynactin distinguished from mutant dynactin?

4) Line 181. Authos stated 'just above backgrounds staining level', but this is not so robust unless your staining is very weak?. Others have use a 1.5 - 2 fold greater than background levels to be sure staining is true and not background noise to avoid reporting false negative data.

5) Line 183-184 and discussion. Is Fas+ cells less than 8 um? that is very smaller or similar to the nucleus size, which suggest it is more a cellular debris rather than a cell. Also, what confirms the positive stained are cells and not debris in 2B?

6) There was no timeline of the age of the phenotypes (e.g. when symptomatic, end stage) as difficult to compare with other ALS models such as with hG93A SOD1.

7) Figure legend 1 and others. There is too much result detail in the figure legend as it repeats information from the result and a very long figure legend.

8) Figure 3. lacks logically organisation in layout and the body text for fig 3 A-F are not in the order of A, B, C etc. Also, lacks WT control images for comparison.

9) Confused why the axonopathy is specifically at this region of the white matter. Would it not be more random? Is this a cutting artifact? Please comment.

10) Line 333-334. How do authors control for fission and fusion of mitochondria? This could cause a higher or lower count, respectively. Also, this would make fig 4F redundant.

11) Fig. 5. Weak staining in panels A-C. Needed a positive control brain region such as the hippocampus or central canal of spinal cord to show positive BrdU signal.

12) Line 405-406. Graph 5H suggest more damaged motor neurons in mdivi than vehicle, but mdivi is supposed to be neuroprotective. Also, how were damaged motor neurons identified?

13) Fig 6. Unconvincing staining. The DAB staining seems under developed in the WT and the mutant is over developed, thus giving a false positive for the mutant due to the high background staining. Cleaved caspase 3 should have a very distinct cytoplasmic and/or nuclear staining only. Similarly, Panel E does not look the same processed tissue as panel F.

14) fig 7. I can see potential positive brown stained cells in panel A top middle region of control tissue. Also, problem with distinguishing the neurons in panel 7B.

15) Fig 8. Unconvinced panel A staining has worked. the background is much lighter than in panel C giving a false negative result. don’t see margination of Nissl substance to indicate chromatolysis in panel 8D. In panel 8E, the figure legend suggests the same cell was labelled with Cresyl violet and nitrated Hsp90. Also missing scale bar size.

16) Lines 502-505. Repeat of above in discussion at 499-502.

17) Lines 509-517. Repeat introduction information.

18) The novelty of the manuscript stated ‘first to show aberrant mitochondrial localisation and morphology in his model’ needs to be highlighted more as it’s buried in the discussion and weakly mentioned in the title.

19) Line 575-583. confused why a section on 3-nitrotyrosine was discussed when not studied in this manuscript.

20) Missing conclusion.

Minor concerns:

1) Line 53. Avoid the term ‘believe’ as should use ‘suggest’ since ‘believe’ is a personal conviction

2) Lines 79, 427. Define Tg, CC3

3) Line 160. Unclear why study cleaved caspase 8 when cleaved caspase 3 was studied.

4) Methods. Missing company and product codes.

5) Line 427. P value too detail. P<0.001 is sufficient.

6) Line 465. CC*?

7) Line 499. Need to italicise gene name

8) Videos were unavailable.

Author Response

Reviewer 3.

  1. This is not the first study of a mutant dynactin model of MND.

Response. We have done a better job at reviewing the literature regarding other mutant dynactin mouse models. We elaborate on these in the Discussion.

  1. ICV injections.

Response. We have clarified the experimental protocol for the ICV injections in the Methods. The volume was delivered not as a single acute injection bolus but as slow infusion over time.

  1. Insufficient to provide western blot of dynactin only in suppl.

Response. This single western blot is not a main aspect of the study and is presented for model validation in brief. Most of the validation of the model has been presented in the Laird et al 2008 paper. We use a FLAG antibody because the human dynactin construct was designed with a FLAG tag so the human dynactin-1 could be distinguished for the endogenous (host) dynactin. There are no antibodies available to specifically detect human dynactin-1 divisible from mouse dynactin-1.

  1. Background staining.

Response. We cleared how the immunohistochemistry was interpreted in the Methods sections and included a better explanation of the immunohistochemistry so that background and debris was not a factor.

  1. Fas cell size.

Response. The fas-positive cells could be seen as clear individual small cells that were about 7-8 microns in diameter (Figure 2). However, the Fas cells could cluster, particularly around degenerations motor neurons. Some are seen in Figure 2B.  These clusters were not included in the counts.

  1.  

Response. We have better articulated a timeline for disease in the Results and Discussion.

  1. Figure legend length.

Response. We have shortened the legened for Figure 2 by about 2 lines.

  1. Figure 3 organization.

Response. Figure 3 layout has been completely reorganized.

  1. Confused why axonopathy is specifically in this region.

Response. Our mouse model has considerable axonopathy in brainstem and spinal cord. This was just one example.

  1. How do authors control for fission and fusion of mitochondria?

Response. Divisible SOD2-positive particles were counted irrespective of fission or fusion. The counts are not weight on oer determined by fission or fusion events.

  1. Figure 5. Weak staining for A-C.

Response. We have included a positive control for BrdU staining in Figure 5A (inset).

  1. More damaged motor neurons in mdivi1-treated mice.

Response. While mdivi-1 protected against the loss of motor neurons, the remaining motor neurons appeared to have residual damage. Thus mdivi-1 might nor be fully protecting the motor neurons or this is due to off-target effects of mdivi-1.

  1. Unconvincing staining.

Response. The spinal sections from WT and mutant mice were processed at identical times using identical reagents in the same trays in replicated experiments. The antibody to cleaved caspase-3 is highly specific and dose not detect any full-length proenzyme. That is the likely explanation for the very low staining in the WT.

The clearly particulate staining for cleaved caspase-3 in the cytoplasm has been seen before in human motor neurons (Martin, 2008). The neuropil staining could likely reflect dendrite or synapse staining.

The control TUNEL image was taken with an incorrect filter. It has been replaced.

  1. Staining in figure 7.

Response. We did see some faint staining for cleaved caspase-8 in WT-DCNT1 mice. The neuron in Figure 7B is now outlined for better clarity.

  1. Figure 8.

Response. All the sections were processed identically at the same time with the same DAB solution for the same times. The experiments were replicated several times. The antibody is highly specific for only nitrated Hsp90-Y56. The control sections do not appear to have pathological nitration of Hsp90 and is not a false negative.

The broad black arrows in Figure 8D identify the chromatolytic neurons.

The legend has been edited for clarity.

Scale bar size is indicated.

 16, 17.  Editing.

Responses test has been edited.

  1. Novelty.

Response. The test has been modified in the Introduction and Discussion to clarify novelty.

  1. Why 3-nitrotyrosine in Discussion.

Response. 3-nitrotyrosine is discussed because we studied 3-nitrotyrosine modified Hsp90.

  1. Conclusion is missing.

Response. A conclusion has been added.

Minor concerns.

  1. Use of believe.

Response. The has been changed.

  1. Define tg, cc3

Response. These have been spelled out at first use.

  1. Use of cleaved caspase-8.

Response. This have been explains because it is upstream of caspase-3 activation and is related to T cell mediated cell death.

  1.  

Response. Company names for supplies are in the methods.

  1. P value.

Response.  P values have changed in some instances.

  1. CC is corrected
  2. Gene reference has been italicized.
  3. Videos are accessible.

Reviewer 4 Report

Comments and Suggestions for Authors

This manuscript employs neuron-specific transgenic mice expressing human DCTN1 (WT or ALS-linked G59S) to map a staged motor-neuron degenerative cascade: axonopathy and chromatolysis; early mitochondrial accumulation and fragmentation progressing to depletion with swelling. The authors also observed activation of cell death alongside perisomatic T-cell apposition; and emergence of nitrated Hsp90. Furthermore, they show that a targeted intervention—Drp1 inhibition with Mdivi-1—partially normalizes mitochondrial pathology, reduces the burden of “damaged” neurons, and extends survival, linking transport defects to fission-driven mitochondrial failure. Overall, the immunohistochemistry is technically clean, anatomically precise, and quantitatively handled in a way that addresses key open questions about how dynactin perturbation in motor neurons connects impaired axonal transport to mitochondrial and cytotoxic signaling. The only major concern I have is how short the introduction is, and it misses out several past works from multiple labs. I encourage the authors to improve their introduction and add relevant citations, so that the readers can comprehend the subsequent results and discussion.

Overall, the work is suitable for publication pending minor revisions.

  1. Typographical error: “DCNT1” should be “DCTN1” (p2:61). Check throughout

“To determine the mechanisms whereby human mutant DCNT1 causes degeneration.”

  1. Figure 1D caption is ambiguous. It states “Some motor neurons in mutant- and wildtype-DCTN1 tg mice were classified as damaged…” but does not specify which genotype(s) the image(s) in panel D represent. Please indicate whether the panel shows WT or G59S (or split into D1/D2). This ambiguity also complicates interpretation alongside Fig. 1H, where the text claims wildtype had more damaged neurons than mutants—please verify labels and axes.

  1. Figure 1H quantification. Please add a panel showing the percentage of damaged motor neurons —damaged/ (damaged + intact)—by genotype, using the same anatomical levels and classification criteria. This normalization will place the current raw counts (number/mm²) in context of the approximate 50% motor-neuron loss in mutants and help resolve the counterintuitive statement that wildtype shows more damaged neurons than mutants.

  1. Figure 5H normalization. Please normalize “damaged motor neurons” to total motor-neuron count per animal and plot this alongside the raw counts. This will control for genotype/treatment differences in total neurons (see Fig. 5G) and allow a fair comparison of injury burden.

  1. Figure 7D legend typos. “CC*-positive motor neurons” should read “CC8-positive motor neurons,” and “motor neuros” should be corrected to “motor neurons.”

  1. Literature context on transport and CAP-Gly MTBD. The manuscript discusses dynactin’s role in retrograde transport and the CAP-Gly (G-rich) MTBD but does not cite foundational transport-activity and plus-end targeting studies on p150 (including G59S). Please add primary work demonstrating reduced cargo initiation and processivity ,and disrupted CAP-Gly/+TIP interactions (e.g., Moughamian 2012; Levy 2006; Moore 2009) where you introduce dynactin’s transport role and reference them again in the Discussion to anchor your phenotypes in established functional evidence.

  1. PINK1–Parkin mitophagy rationale. Given the staged mitochondrial pathology in mutants—early perikaryal accumulation/fragmentation followed by depletion with swelling—and the benefit of Drp1 inhibition, assessing PINK1–Parkin mitophagy would substantially strengthen the mechanism. Please evaluate whether mitophagy is activated, impaired, or mistimed in motor neurons (e.g., Parkin recruitment). This would mechanistically bridge the transport defect to mitochondrial quality-control outcomes.

Author Response

Reviewer 4.

  1. DCTN typos

Response. The Typos have been corrected.

  1. Figure 1 D legend.

Response. The legend has been edited for clarity.

  1. Figure 1H quantification.

Response. A graph has been added to Figure 1 showing the normalization of damaged motor neuron to total motor neurons.

  1. Figure 5 normalization.

Response. A graph has been added (Supplementary Figure 4) showing the normalization of damaged motor neurons to total motor neurons.

  1. Figure 7 legend typos.

Response. Typos have been corrected.

  1. Literature missing.

Response. We have augmented this literature and have cited all of the references suggested.

  1. Mitophagy

Response. We have done new experiments to address the mitophagy question. New immunohistochemistry was done for Parkin. The results are present in a new figure (Supplementary Figure 3).

Round 2

Reviewer 2 Report

Comments and Suggestions for Authors

I appreciate the authors’ response and the additional experimental data, which have improved the manuscript. I have no further questions.

Reviewer 3 Report

Comments and Suggestions for Authors

All concerns have been addressed. No further comment.